# Mapping of Mean Deformation Rates Based on APS-Corrected InSAR Data Using Unsupervised Clustering Algorithms

**Mohammad Amin Khalili** [1,*] **, Behzad Voosoghi** [2] **, Luigi Guerriero** [1] **, Saeid Haji-Aghajany** [2,3] **, Domenico Calcaterra** [1] **and Diego Di Martire** [1]

1   Department of Earth, Environmental and Resource Sciences, Monte Sant'Angelo Campus, Federico II University of Naples, 80126 Naples, Italy
2   Faculty of Geodesy and Geomatics Engineering, K. N. Toosi University of Technology, Tehran 15433-19967, Iran
3   Institute of Geodesy and Geoinformatics, Wrocław University of Environmental and Life Sciences, Norwida 25, 50-375 Wrocław, Poland
*   Correspondence: mohammadamin.khalili@unina.it; Tel.: +39-34-4413-7462

**Abstract:** Different interferometric approaches have been developed over the past few decades to process SAR data and recover surface deformation, and each approach has advantages and limitations. Finding an accurate and reliable interval for preparing mean deformation rate maps (MDRMs) remains challenging. The primary purpose of this paper is to implement an application consisting of three unsupervised clustering algorithms (UCAs) for determining the best interval from SAR-derived deformation data, which can be used to interpret long-term deformation processes, such as subsidence, and identify displacement patterns. Considering Port Harcourt (in the Niger Delta) as the study area, it was essential to remove the sources of error in extracting deformation signals from SAR data, spatially ionospheric and tropospheric delays, before using UCAs to obtain its characteristics and real deformation data. Moreover, another purpose of this paper is to implement the advanced integration method (AIM) for atmospheric phase screen (APS) correction to enhance deformation signals obtained through different SAR processing approaches, including interferometric SARs (two-pass interferometry, InSAR) and multitemporal interferometry SARs (n-pass interferometry, DInSAR; permanent scatterer interferometry (PSI); and small baseline subset (SBAS)). Two methods were chosen to evaluate and find the best technique with which to create an MDRM: The first one was to compare the signals corrected by the AIM and the vertical component of the GPS station, which showed the AIM providing 58%, 42%, and 28% of the matching with GNSS station outputs for InSAR, PSI, and SBAS, respectively. Secondly, similarity measures and Davies–Bouldin index scores were implemented to find an accurate and reliable interval in which the SBAS technique with the unsupervised K-medians method has been chosen. Based on GNSS vertical deformation in a 500 m radius around the station, the SBAS K-medians technique expressed up to 5.5% better deformation patterns than the map of SAR processing techniques.

**Keywords:** interferometric SAR; multitemporal interferometry SAR; advanced integrated method; unsupervised clustering algorithms



## 1. Introduction

Earthquakes, volcanoes, the movement of tectonic plates, and changing groundwater levels lead to the vertical deformation of the ground, known as subsidence and uplift. Changing groundwater levels due to natural and/or anthropogenic factors can cause subsidence of tens of meters in some parts of the world [1–7]. Groundwater levels are controlled by various natural factors, such as rainfall, temperature, and human activities, such as extensive groundwater harvesting for agricultural applications. Therefore, the study of deformation is a tool with which to monitor the number of changes in groundwater storage and can help maintain these resources, prevent drought, and analyze land subsidence

or uplift around large cities caused by groundwater extraction [8–11]. Global challenges, such as disaster prevention, land-use monitoring, and climate change, can be addressed by developing remote sensing and automatic Earth-monitoring technologies. Due to advanced remote sensing, various techniques have been developed to study the deformation caused by subsidence, including global navigation satellite systems (GNSS) and interferometric synthetic aperture radar [12].

The latter is an appropriate technique for monitoring vertical surface deformation with millimeter accuracy. It can provide a high level of spatial resolution for determining the deformation rate over a vast area [13]. Since the launch of this technology, several techniques have been developed to provide increasingly accurate results. These techniques span from the interferometric SAR (two-pass interferometry, InSAR) [14] to the much more sophisticated multitemporal interferometry SAR (MTInSAR) techniques, such as persistent scatterer interferometry (PSI) [15], and the differential principle imbued with time series analyses such as small baseline subset (SBAS) [16]. Each has benefits and drawbacks in retrieving short- and long-spanning spatiotemporal aspects of surface deformation.

The main observation of InSAR includes the deformation signal and the signal caused by topographical change, orbital errors, atmospheric effects, and diffusion changes. Therefore, according to the study area and which above delays have more effects on the deformation signal obtained from InSAR, utilizing proper corrections is compulsory. In this case study, Port Harcourt is located near the equator and next to the Atlantic Ocean. Due to its hot and humid climate, the amount of water vapor in the troposphere and its temporal changes are significant. Indeed, to find the real signal of deformations, atmospheric phase screen (APS) correction is necessary, which removes and decreases the spatial and temporal decorrelation of InSAR interferograms [17]. Studies that considered the effect of APS on InSAR deformation [18–20] mainly used a single InSAR processing scheme (for example, the MTInSAR technique); however, using any InSAR processing technique leads to different results when recovering small and large deformations. Indeed, the conventional InSAR technique can recover large deformations that occurred in a relatively short period, but has the greatest decorrelation due to the atmosphere and other effects, while MTInSAR methods minimize the decorrelation but have limitations in detecting significant deformation signals due to the selection of coherent pixels according to a prior deformation model [21–24]. One of the new APS correction methods has been proven successful and implemented into the SBAS technique [25], namely the advanced integration method (AIM), which, in this study, is implemented for the first time in three different techniques (InSAR, PSI, and SBAS) and on a new case study.

Unlabeled remote sensing data make supervised learning algorithms ineffective. An unsupervised learning algorithm can be a means of discovering hidden patterns in data [26]. Clustering is one of the most important unsupervised learning problems. Therefore, as with every other problem, it involves identifying a structure in a collection of unlabeled data. Unsupervised clustering is a technique for finding the internal grouping in an unlabeled dataset [27]. Some papers were worked on to improve and use unsupervised clustering [28–31]; unsupervised CNN learning methods were used for remote sensing image representation and scene classification [32]; marsh elevation and landcover evolution for coastal management was implemented by a K-means unsupervised algorithm [33]; image segmentation can be performed to assort all pixels in groups by K-means clustering [34]; and unsupervised deep learning methods were used to generate a landscape typology for Switzerland [35].

In this study, an application incorporating the APS correction method, which utilizes ERA5 meteorological data (AIM), will be developed for both InSAR and MTInSAR processing techniques (PSI and SBAS) to assess the value of incorporating such corrections into SAR image processing. The concept of using UCAs to determine the accurate, reliable, and best interval is also extended for monitoring straightforward MDRMs and finding the zones of risk to better decipher displacement distribution patterns as well as identify

the best processing strategy via comparing the data derived from the GNSS and GRACE satellite [36].

## 2. Case Study

The study area comprises the town of Port Harcourt, located in the Niger Delta basin. This basin extends over the coastal and oceanward sectors of the much larger and older Benue Trough in addition to an NE–SW folded rift basin that runs diagonally across Nigeria. It formed simultaneously with the opening of the Gulf of Guinea and the equatorial Atlantic in Aptian–Albian times when the equatorial part of Africa and South America began to separate. The structural evolution of the Niger delta was influenced by the movements along the equatorial Atlantic oceanic fracture zones that extended beneath the delta and determined the initial locus into which the proto-Niger built its delta.

Port Harcourt is the capital city of the southern Nigerian state of Rivers (Figure 1). This town is located along the Bonny River (an eastern tributary of the Niger River), 41 miles (66 km) upstream from the Gulf of Guinea [37].

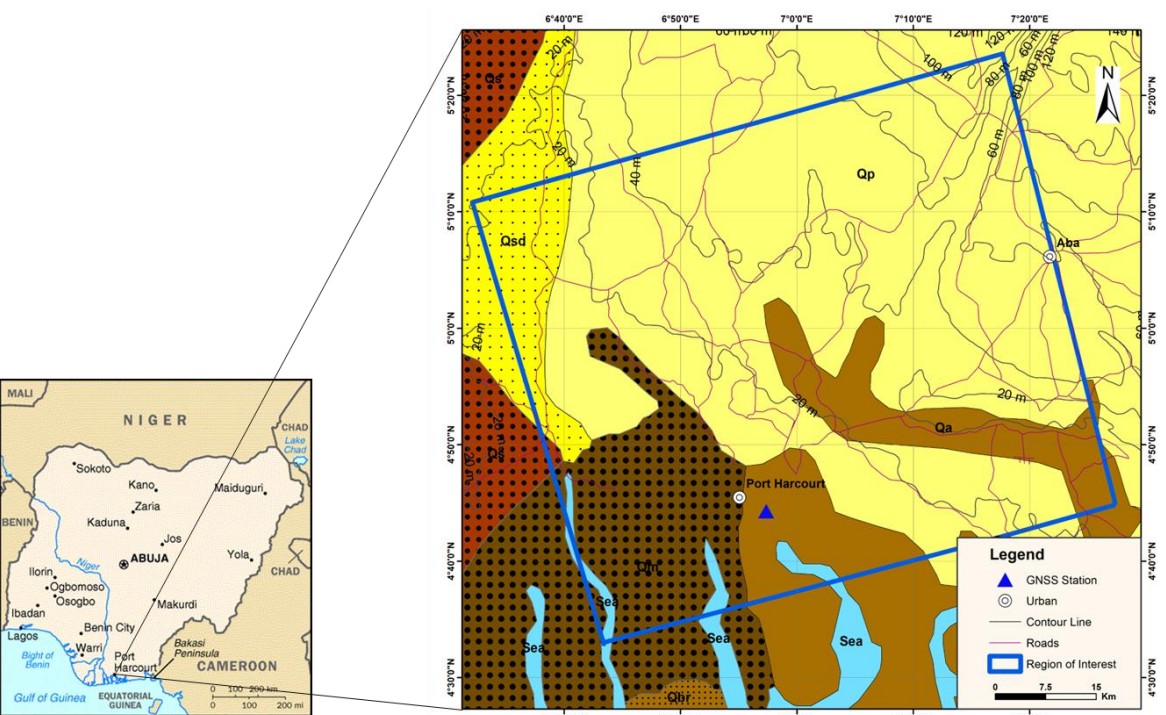

**Figure 1.** Geological map of Nigeria, showing the basement complex, sedimentary basins, and the study location (Port Harcourt). Tm, Paleocene Imo Formation; Tb, Ameki Group; Tl, Og washi Asaba Formation; and Qp, Benin Formation.

From a geological point of view, the Port Harcourt area is underlain by sedimentary rocks of the Niger Delta. These rocks primarily consist of fluvial deposits of the Benin Formation (Qp) and related deltaic plain deposits (Qsd), meander belt deposits (Qs), mangrove swamp deposits (Qm), abandoned beach ridge deposits (Qbr) and recent alluvium (Qa) [38,39].

The rifting process responsible for forming the Benue Trough also created the Port Harcourt sedimentary basin. This basin is structurally oriented in an NNW–SWS direction, and it stretches over 800 km while being 150 km wide. More than 6000 m of sediments is dated to the Cretaceous–Tertiary, and those dated before the mid-Santonian are uplifted, faulted, and deformed extensively (Figure 1) [40].

One of the primary aquifers in this area consists of a multi-aquifer system [41]. It also consists of a deep aquifer, and the water aquifer is approximately 750 m deep. The

thickness of the aquifer and the percentage of available sand increase from the north to the south of the region [41].

There is a substantial relationship between the amount of and the temporal variation in water vapor in the troposphere, as mentioned in the introduction. As a result, the displacement fields obtained from SAR processing approaches are impacted by the remarkable atmospheric phase and consequently affect the displacement signal. For a better review, the relative humidity obtained from ERA5 data in the center of the area from 2015 to 2020 is shown in Figure 2, where seasonal fluctuations in relative humidity can be seen. Among the indices in estimating the temporal changes in tropospheric delay, relative humidity is more important due to more temporal variations and fluctuations.

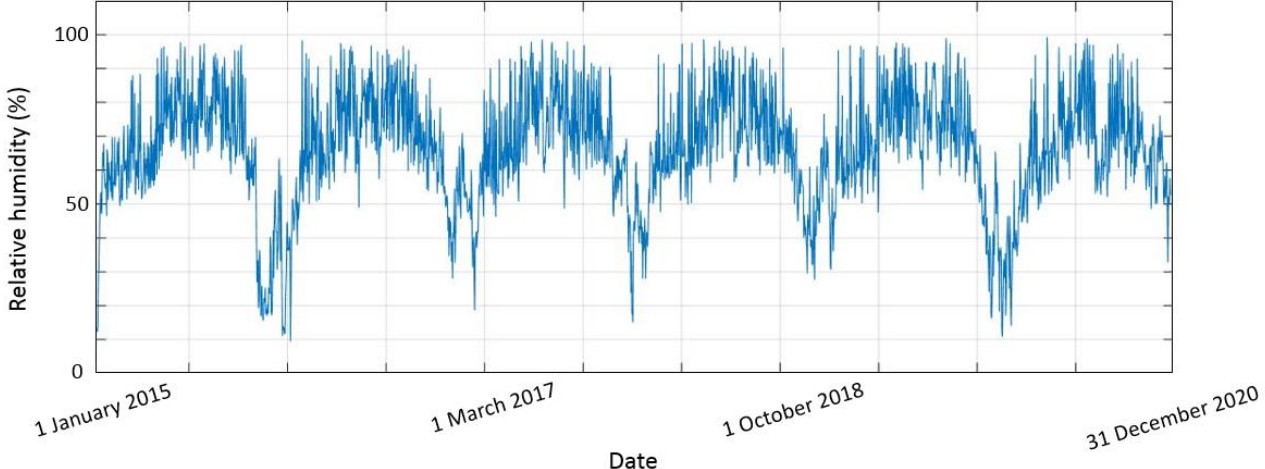

**Figure 2.** Relative humidity in the center of the study region.

Figure 3 illustrates a GNSS station's (coordinates: long, 6.9°; lat: 4.6°) vertical component time series for 2015–2017 to better understand the area's subsidence. As can be seen, the fluctuations in the vertical component due to seasonal changes are pretty evident. In addition to seasonal fluctuations, the downward trend of this chart can also be seen. A line has been fitted to the time series to assist in obtaining a better conclusion. It can be seen from the equation of this line that the time series of the vertical component indicates the occurrence of subsidence at the location of the GNSS station.

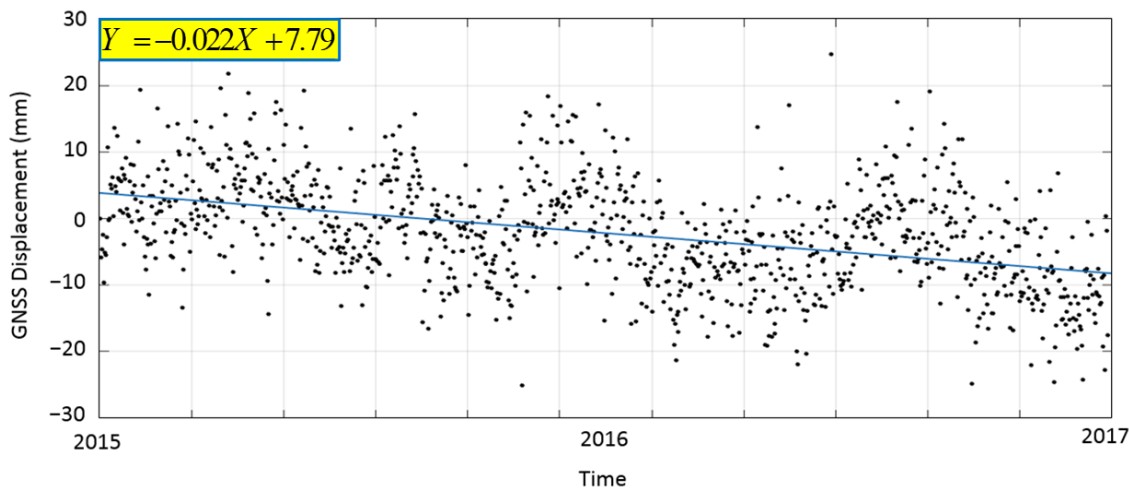

**Figure 3.** Time series of vertical components for a selected GNSS station and the linear fitting.

### 3. Datasets

Sentinel missions are parts of the European Union's Earth observation program, named Copernicus, and are managed by the European Commission in cooperation with the European Space Agency (ESA). This satellite's revisiting is 12 days, and data on the desired area is available every six days. In this study, Sentinel-1A radar acquisitions from track 103 (ascending) between 2015 and 2017 (37 images) have been used to perform SAR processing approaches (InSAR, PSI, and SBAS). The complete characteristics of the acquisitions can be seen in Table 1.

**Table 1.** Specifications of the radar acquisitions.

| Mission | Acquisition Time (Images) | Product | Path | Swath | Pass | Central Look Angle (Deg) | Polarization |
|---|---|---|---|---|---|---|---|
| Sentinel-1A | 2015 to 2017 (37) | Single-look complex (SLC) | 103 | IW TOPS | Ascending | 38° | VV |

In this study, ERA5 reanalysis data published by the European Centre for Medium-Range Weather Forecasts (ECMWF) are used to implement the advanced integration technique. The ERA5 reanalysis data constitutes a climate reanalysis dataset covering 1950 to the present. This model published values of meteorological data, including water vapor pressure and temperature for 37 pressure levels [42].

This study uses two different strategies to evaluate the obtained results. The first method uses the calculated displacements from observations of GNSS stations in the study area. The second is based on the GRACE satellite observations of groundwater storage level changes.

GRACE satellite observations provide terrestrial water storage (TWS) data [43,44]. In 2002, the German Aerospace Centre (DLR) as well as the National Aeronautics and Space Administration (NASA) launched the GRACE mission at a height of about 500 km. The project presented Earth's gravity field at a high resolution and with high accuracy. Numerous pieces of research have proven that the change in groundwater storage based on Earth's mass change and the same mass concentrated over a small region can be estimated by GRACE observations [43,44]. In this paper, the monthly Level-2 gravity field products from Release-5 (RL05), up to a spherical harmonic coefficient of 60 degrees, are used to compute land subsidence due to groundwater depletion. More details about this process can be found in previous research [45].

### 4. Methods

As a result of the use of SAR data, sources of error must be eliminated, and the decorrelation of deformation signals must be significantly reduced based on the characteristics of the study area. Considering the study area, an investigation has been carried out to dispose of tropospheric and ionospheric delays. The deformed zones were also interpreted and improved by using three UCAs, which were used to divide the intervals and express the deformation maps accurately. The purpose of this section of the paper is to introduce the methodology in three steps. To begin with, interferometry SAR (InSAR), PSI, and SBAS produce results (velocity). Secondly, the AIM is applied to the interferograms obtained from all of the processing approaches to remove tropospheric delays. The results and improvements are compared with the vertical component of the GNSS signal. Furthermore, combining processing approaches and UCAs will yield the most accurate and reliable maps of deformation and risk areas, in addition to choosing the best technique for SAR processing. Figure 4 illustrates all of the processing stages in this study.

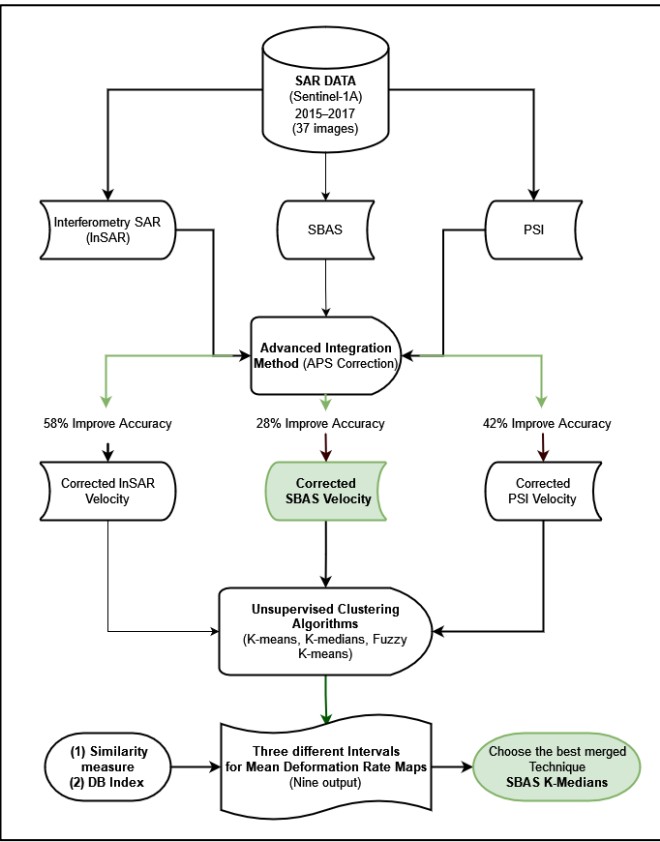

**Figure 4.** Flowchart.

### 4.1. Interferometric Processing

The InSAR method is based on signal extraction related to discrimination from SAR interferogram images. By obtaining the phase difference recorded at corresponding pixels in two successive SAR images obtained for a region at two different times, an interferogram is formed, which is the main observation of InSAR. The differences in the phases obtained in the interferogram include the deformation signal and the signal caused by topographical change, orbital errors, atmospheric effects, and diffusion changes. More precise details are given in [46]. For solving limitations such as decorrelation, phase unwrapping, and atmospheric delay, multi-temporal interferometry (MTI) [47] using advanced synthetic aperture radar differential interferometry (A-DInSAR) has been investigated for monitoring subsidence-induced deformation. Small baseline subset (SBAS) and permanent scatterer interferometry (PSI) are coherence-based A-DInSAR methodologies. These techniques can measure ground surface deformations accurately, spatially densely, and across a large area [15].

#### 4.1.1. Interferometric SAR (Two-Pass Interferometry, InSAR)

The InSAR method is the classic and primary approach in InSAR analyses for recovering the deformation of a region. This technique can be applied to two SAR images obtained from a region (before and after deformation) to investigate deformation behavior [46].

#### 4.1.2. PSI Method

By utilizing the PSI technique, radar targets (known as permanent scatterers, or PSs) are virtually immune from geometric as well as temporal interferometric effects and demonstrate high interferometric coherence because of their remarkably high stability over time. PSs manifest as objects already existing on the ground, such as artifacts (buildings, monuments, roads, railways, and pylons) or natural elements (rocky outcrops). In contrast, vegetated or occasionally snow-covered areas lack PSs. It is possible to analyze all of the

interferograms contained in a stack of images by using cross-correlation and applying that analysis to a single pixel with a particular level of quality or stability. In this study, minimum coherence is considered 0.5, and the phase threshold is assumed to be 0.005 [47,48].

### 4.1.3. SBAS Method

The SBAS technique is another MTI accurate processing technique based on time series analyses. This technique selects pairs of images with the minimum temporal and spatial baselines from time series interferograms. As a result of this approach, the temporal and spatial decorrelations are minimized. As with the PSI method, coherent pixels in the time series of images are identified using a primary deformation model. With many interferograms available over a relatively long period in addition to non-linear modeling, each atmospheric, orbital, deformation, and topography effect can be separated into selected, coherent pixels [16]. In this study, for processing SAR images based on SBAS, the minimum coherence is considered to be 0.5, and the phase threshold is equal to 0.005.

### 4.2. Atmospheric Correction on InSAR Signals

The atmospheric phase screen (APS) is one of the primary sources of error in deformation recovery using SAR processing approaches. This technique is based on the effects of the ionosphere and troposphere on atmospheric signals. Generally, the effect of the ionosphere layer on short-wavelength InSAR is negligible [49]. The main cause of this error is the variations in the water vapor density, temperature, and pressure of the troposphere layer. Based on previous research, it can be said that variations in temperature and pressure are not effective enough to generate a localized phase gradient in InSAR observations [50].

The APS can be computed by using known atmospheric models, and can be removed from observations or estimated by InSAR temporal and spatial data. The advanced integration method (AIM) is used in this study to mitigate this error.

#### Tropospheric Correction Using the AIM

An InSAR analysis is performed by using the time delay and differential phase shift of radar acquisitions. When the radar signals pass through the troposphere layer the velocity decreases, and the observations are affected due to spatially variable tropospheric [51] delays. The zenith tropospheric path delay (ZTD) is obtained from the numerical integration between the surface elevation ($z_0$) and the troposphere top layer ($z$) [51]:

$$\text{Delay\_ZTD} = 10^{-6} \left\{ \frac{k_1 R_d}{g_m} P(z_0) + \int_{z_0}^{z} \left( (k_2 - \frac{R_d}{R_v} k_1) \frac{e}{T} + k_3 \frac{e}{T^2} \right) dz \right\} \quad (1)$$

$$k_1 = 0.776 \text{ KPa}^{-1}, \quad k_2 = 0.716 \text{ KPa}^{-1}, \quad k_3 = 3.75 \times 10^3 \text{ K}^2\text{Pa}^{-1}$$

where $R_v$ (461.495 J/kg/K) and $R_d$ (287.05 J/kg/K) are the specific gas constants for water vapor and dry air, respectively; $e$ is the water vapor pressure; $P(z_0)$ is the surface pressure; $T$ is the temperature in K; and $g_m$ is the gravitational acceleration, $g$, averaged over the troposphere layer.

According to previous research, it is necessary to use mapping functions to convert the ZTD into the LOS direction. For example, the following equation can estimate the tropospheric delay in the LOS direction by using a simple mapping function [25,45,51]:

$$\text{Delay\_LOS} = \frac{\text{Delay\_ZTD}}{cos(\theta)} \quad (2)$$

where $\theta$ is the incidence angle of the radar signal. The use of mapping functions leads to an error in the amount of computational tropospheric delay. Therefore, in this paper, direct numerical integration on the LOS has been applied instead of integrating the zenith direction and mapping functions to project the obtained delay on the LOS. This method is called the advanced integration method (AIM). Figure 5 shows a schematic image of the

difference between the standard and AIM methods. It is possible to predict tropospheric parameters along the satellite LOS direction by using gridded global atmospheric model products. Several times a day, global atmospheric models provide three-dimensional gridded tropospheric parameters at different pressure levels. ECMWF ERA5 reanalysis products are provided hourly at 0.25° spatial grids with 37 atmospheric pressure levels. The ERA5 outputs that extend two degrees beyond the coverage of the SAR images that we intend to correct in all directions are downloaded. Considering the fact that the most significant time difference between the SAR acquisitions and the ERA5 outputs would be up to 30 min, we linearly interpolate the ERA5 outputs based on the two closest ERA5 outputs when the time difference is greater than 20 min. Unless otherwise specified, we use the nearest ERA5 output. Equation 2 calculates the satellite LOS delay using these parameters [25].

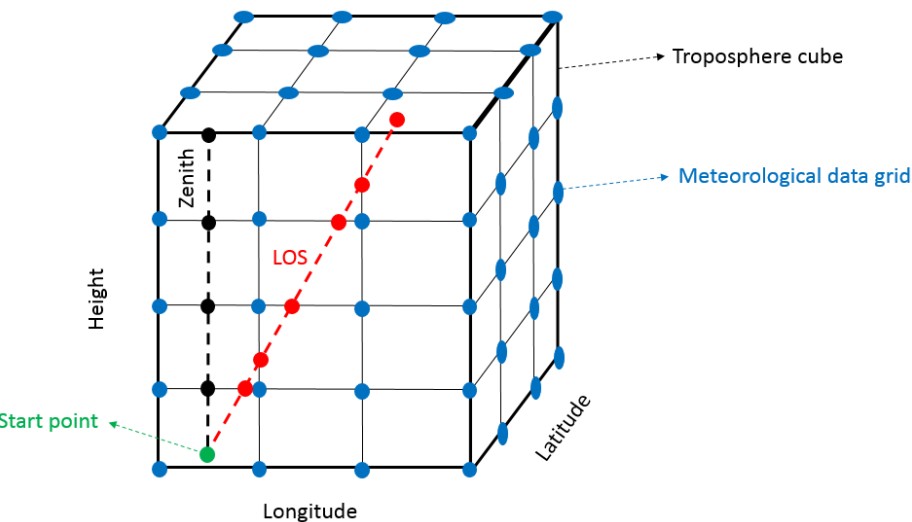

**Figure 5.** Difference between integration on the zenith and LOS directions.

The mentioned equations and Figure 5 show that implementing these methods requires a 3D meteorological grid, and meteorological data on the signal path need to be calculated using interpolation algorithms. The AIM also needs to compute the distance traveled by the signal in each vertical layer of the meteorological grid. More information can be found in [25], which used this method on only the SBAS processing technique, and this study concentrated on implementing this method in three different processing approaches.

The performance of converting the zenith into the LOS direction, as well as of the AIM, was evaluated based on Sentinel-1A SAR datasets. A case study was conducted to validate the applicability of the corrections by studying SAR images acquired under complex weather conditions over different seasons. Differential interferograms were generated using SARscape software created by L3Harris, SARmap SA, and United States Geological Survey digital elevation models to remove topographic phase components. Since values in a wrapped interferogram are limited to $-\pi$ and $\pi$, a phase elevation trend cannot be detected using wrapped phases. Therefore, unwrapped phases were used after solving phase ambiguities. All differential interferograms were unwrapped using the SNAPHU package [52].

It should be noted that the required meteorological parameters are accessible by using different published data [51,53]. Moreover, it is necessary to use this APS correction to find reliable and accurate data with which to prepare detailed and trustworthy mean deformation rate maps with sufficient information.

*4.3. Data Classification*

In many fields, including remote sensing, data classification is used to perform complex, varied tasks and use data to maximize efficiency and effectiveness. Applying an

appropriate classification method can lead to accurate MDRMs, which provide an idea of what is happening. Preparing detailed and accurate deformation maps is much more suitable and straightforward for interpreting phenomena (in this case, subsidence). In the following parts, three UCAs are discussed using the APS-corrected output of three different SAR processing approaches.

Feature-based unsupervised clustering algorithms are multivariate, which means that they have more than one variable. In this study, the longitudes and latitudes of points, along with the velocities obtained from each processing approach, are entered into these algorithms as the three main and variable features. In unsupervised clustering, no labels are given to the clusters. Based on elbow tests and knowledge from the study area, 5 clusters were selected to implement these algorithms. In fact, we have rows with the number of DSs and PSs, which are created from columns with longitude, latitude, and velocity, and the purpose of this clustering is to create a new column with a label from 0 to 4 in the obtained datasets which come from APS-corrected data. During the implementation of unsupervised clustering, five clusters are introduced to the algorithms, and the centroid for each cluster is selected according to the dataset available for the algorithms. Each point or row is then entered into the algorithm individually and measured with different distances for each clustering algorithm. Ultimately, they will be placed in a proper cluster after being updated five times. After the three clustering methods have been fully implemented, passing through a proc. is in the form of the union–intersection principle and is proportional to the clusters. For each UCA, the number of points and their changes provide an appropriate interval. In QGIS software, mean deformation maps are drawn more accurately and reliably based on these algorithms' interval outputs. Before this, experts trusted the intervals provided by software or processing packages, which are presented in a general form and based on conventional methods such as equal intervals and standard deviations; it can be said that the displacement maps are not well-displayed and ultimately cannot be interpreted.

### 4.3.1. Unsupervised Clustering

Unsupervised learning aims to discover hidden patterns in data. The learning algorithm is not given labels, leaving it to figure out the structure independently [26]. A clustering technique is used to identify the internal grouping of a dataset without labels. Clustering refers to arranging objects into groups whose members have similar characteristics. Accordingly, a cluster is a group that is "similar" among itself and "different" from objects belonging to other groups.

Clustering algorithms may be classified as listed below [27]:

1. Exclusive clustering.
2. Overlapping clustering.

In the first case, data are grouped exclusively, such that if a specific data point belongs to a definite cluster, it cannot be included in another cluster.

On the contrary, the second type, overlapping clustering, uses fuzzy sets to cluster data, such that each point belongs to two or more clusters with different degrees of membership. In this case, data will be associated with an appropriate membership value.

Three of the most used clustering algorithms are stated below:

3. K-means.
4. Fuzzy K-means.
5. K-medians.

While K-means and K-medians are exclusive clustering algorithms, fuzzy K-means is an overlapping clustering algorithm.

### K-Means Clustering

K-means is one of the simplest unsupervised learning algorithms that solve the well-known clustering problem. The procedure follows a simple and easy way to classify a given

dataset through a certain number of clusters (assume k clusters; in this case, 5 clusters). The main idea is to define k centers, one for each cluster. These centroids should be placed smartly because different locations cause different results. Therefore, the better choice is to place them far away from each other as much as possible. The next step is to take each point belonging to a given dataset and associate it with the nearest centroid. The first step is completed when no point is pending, and an early groupage is done. At this point, we need to re-calculate k new centroids as the barycenter of the clusters resulting from the previous step. A loop has been generated. As a result of this loop, we may notice that the k centroids change their location step-by-step until no more changes are made [26]. The objective function is shown below:

$$J = \sum_{j=1}^{k} \sum_{i=1}^{n} \left\| x_i^{(j)} - c_j \right\|^2 \tag{3}$$

where

$$\left\| x_i^{(j)} - c_j \right\|^2 \tag{4}$$

is a chosen distance measure between a data point, $x_i$, and the cluster center, $c_j$, is an indicator of the distance of the $n$ data points from their respective cluster centers.

K-means is a simple algorithm that has been adapted to many problem domains. The k-means procedure can be viewed as a greedy algorithm for partitioning the n samples into k clusters to minimize the sum of the squared distances to the cluster centers. Although it can be proven that the procedure will permanently terminate, the k-means algorithm does not necessarily find the most optimal configuration corresponding to the global objective function minimum. The algorithm is also significantly sensitive to the initial randomly selected cluster centers. The k-means algorithm can be run multiple times to reduce this effect.

All methods are generally classified into 5 clusters based on elbow tests and the knowledge available in the case study.

K-Medians Clustering

K-medians is another clustering and unsupervised algorithm. Cluster centroids, the vectors located at the center of each cluster, are initialized randomly by the algorithm. Initializing cluster centroids involves generating random points from the dataset [54].

Clusters are then assigned to each instance in the dataset. Using the cluster centroids as reference points, we calculate the distance between each instance and each cluster centroid. Based on this distance, the instance is assigned to the cluster that has the closest cluster centroid. K-medians makes use of the Manhattan distance.

Manhattan distance:
$$\| p - q \|_1 = \sum_{i=1}^{n} |p_i - q_i| \tag{5}$$

where $p$ and $q$ are vectors representing the dataset's instances.

After each instance has been assigned to a cluster, the cluster centroids are shifted. Through this algorithm, the cluster centroid moves to the median value of the dimension of all of the instances assigned to the cluster. Clusters have now been formed. Despite this, there is no guarantee that the clusters will partition the data appropriately. This is because the cluster centroids were randomly initialized. For this reason, the K-medians algorithm is reinitialized repeatedly, saving the best cluster centroids after all iterations.

To quantify how well the clusters formed in each iteration perform, the following cost function is used:
$$J = \sum_{j=1}^{K} \sum_{i=1}^{n} |p_i - \mu_j| \tag{6}$$

where $K$ is the number of clusters and $\mu_j$ is the median vector for cluster $j$.

Cluster centroids are randomly initialized within a loop and then calculated for the specified number of iterations. The cluster centroids with the lowest associated cost function values are selected as soon as the necessary iterations have been completed.

Fuzzy K-Means Clustering

In fuzzy clustering, each point has a probability of belonging to each cluster rather than entirely belonging to just one cluster, as in conventional k-means. Fuzzy k-means explicitly attempts to deal with the problem where points are somewhat in-between centers or otherwise ambiguous by replacing distance with probability, which could be some function of distance, such as having probability be relative to the inverse of the distance. Fuzzy k-means uses a weighted centroid based on these probabilities. The processes of initialization, iteration, and termination are the same as the ones used in k-means. The resulting clusters are best analyzed as probabilistic distributions rather than as challenging assignments of labels. One should realize that k-means is a particular case of fuzzy k-means where the probability function is simply 1 if the data point is closest to a centroid and 0 otherwise [55].

On the other hand, in the fuzzy k-means approach the same given data point does not belong exclusively to a well-defined cluster, but it can be placed in a middle way.

### 4.4. Evaluation Methods of Clustering Algorithms

Cluster evaluation is a necessary step in UCAs. It has been applied in many domains to identify similarities or clusters in unlabeled data. Its performance, however, depends on the characteristics of the data to which it is applied. Since there is no universally best clustering evaluation technique, several are available with varying performance characteristics. In this study, the Davies–Bouldin index (DBI) is used to evaluate unsupervised clustering. Moreover, modified similarity measures are used for UCAs to obtain the accuracy of each method on SAR data. Additionally, following the selection of the best processing technique and the most reliable MDRM, the effect of atmospheric correction on each technique will be discussed in the Section 5, as will its effect on the GPS station, to complete the evaluation of this study.

#### 4.4.1. Davies–Boldin Index

The Davies–Bouldin index is based on the principle of with- and between cluster distances. It is commonly used for deciding the number of clusters in which data points should be labeled. It should be small, so the main motive is to decrease the DB index (DBI) [26]. The lower the average similarity is, the better the clusters are separated and the better the result of the clustering performed.

The Davies–Bouldin index can be calculated as follows:

$$\overline{R} = \frac{1}{N} \sum_{i=1}^{N} R_i \tag{7}$$

The best choice for clusters is where the average similarity is minimized. Therefore, a smaller $\overline{R}$ represents better-defined clusters. More details can be found in [26].

#### 4.4.2. Similarity Measure

A similarity measure is a measurement criterion that is defined based on distance. Both supervised and unsupervised clustering methods require similarity measures to determine how similar objects are. Investigators usually use cluster analysis methods and dissimilarity

measures in various situations. Pearson's correlation coefficient measures the similarity between two objects [56]. Pearson similarity is defined as follows:

$$r(x,y) = \frac{\sum\limits_{i=1}^{d} (x_i - \overline{x})(y_i - \overline{y})}{\sqrt{\sum\limits_{i=1}^{d} (x_i - \overline{x})^2} \sqrt{\sum\limits_{i=1}^{d} (y_i - \overline{y})^2}} \tag{8}$$

where $d$ is dataset size (number of rows of each processing approach), $x_i$ and $y_i$ are the individual training and test sets indexed with $i$, respectively, and $\overline{x}$ in addition to $\overline{y}$ are the mean values of the training and test sets for each cluster, respectively.

This method divides the data from different processing techniques into 20% test sets and 80% training sets. Based on the similarity measure and the correlation distance between each cluster in the training sets, the distance between each member of the test sets is calculated. We then find the cluster with the minimum distance from the total distances of each test set with clusters 0, 1, 2, 3, and 4, and label it to that member of the test set. The obtained results should be compared with the labeling results, and the best method should be chosen based on the standard deviation for all of the test members.

Pearson similarity calculates the correlation between two objects, concerning all attribute values. Correlation considers two objects to be similar if their attributes are highly correlated, even though the observed values may be far apart in regard to the Minkowski distance. Correlation distances have greater significance and value than Euclidean or Manhattan distances since they also consider the linear relationship between the clustered datasets and do not depend exclusively on the values of the clustered datasets.

### 4.5. Satellite Observation Evaluation

Considering that only one GPS station is located in this study area, the deformation trend of the GPS station from 2015 to 2017 was compared to the deformation trend obtained from the SBAS and PSI techniques to evaluate the result. Additionally, GRACE satellite data will be used to confirm the deformations derived from SAR data processing by comparing the water level elevation trend with the GNSS station's vertical component in the study area.

## 5. Results and Validation

### 5.1. Subsidence Signals from InSAR Data

As mentioned, Sentinel-1A radar acquisitions have been applied to interferometric processes utilizing different approaches, including InSAR, PSI, and SBAS. To estimate the mean displacement rate, intrusive effects must be removed from the obtained interferograms. After eliminating these effects by using a digital elevation model and orbital data, interferograms only included tropospheric effects and the deformation phase. The calculated APS based on the AIM must be applied to the raw interferograms. The following samples of obtained APSs from this approach can be seen in Figure 6. In the left and right maps some areas are white, while others are dark and light blue, which indicates the method's accuracy and high spatial resolution. These two figures were created at two different times on different interferograms to demonstrate that the AIM can correct tropospheric delays with high spatial resolution. Additionally, this figure illustrates that unexplained tropospheric effects in interferograms can result in unrealistic displacement signals and incorrect numerical subsidence calculations.

The results from different SAR processing approaches are computed by applying a tropospheric correction method (AIM). The MDRM of the study area was then calculated by using PSI and SBAS algorithms. In addition, the deformation map of the area in the period of the images was obtained by using the InSAR technique. The final interferograms are expected to include only the deformation phase at this stage.

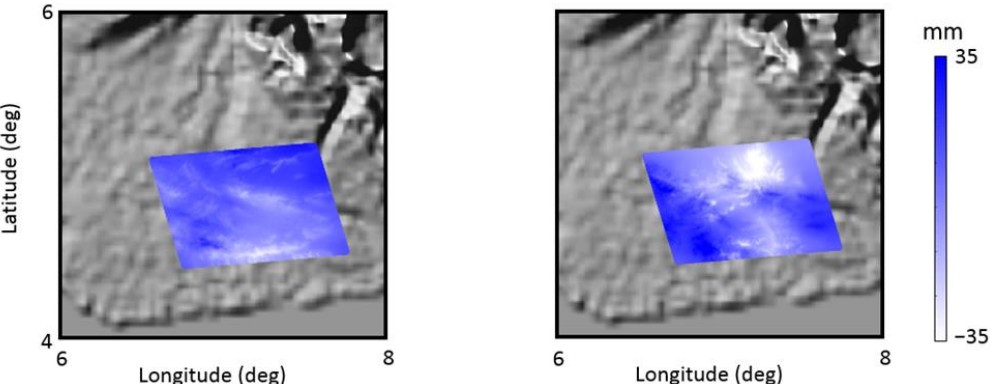

**Figure 6.** Samples of obtained APSs using the advanced integration method on different interferograms. (**Left**): 17 April 2015; (**right**): 19 January 2016.

Using the AIM eliminated incorrect signals. For this reason, the obtained displacement map from the InSAR approach has been converted into a displacement velocity map using the time index. Figure 7 shows the obtained displacement velocity field using various approaches. The upper portion of the figure (a′, b′, and c′) illustrates the effect of ionospheric and tropospheric delays on deformation signals. Atmospheric delays in the study area have significantly altered MDRMs. Conversely, the AIM has been applied to each of the processing approaches. After APS correction and the elimination of tropospheric delays from deformation signals, MDRMs with different deformation zones and velocities (a″, b″, and c″) were generated.

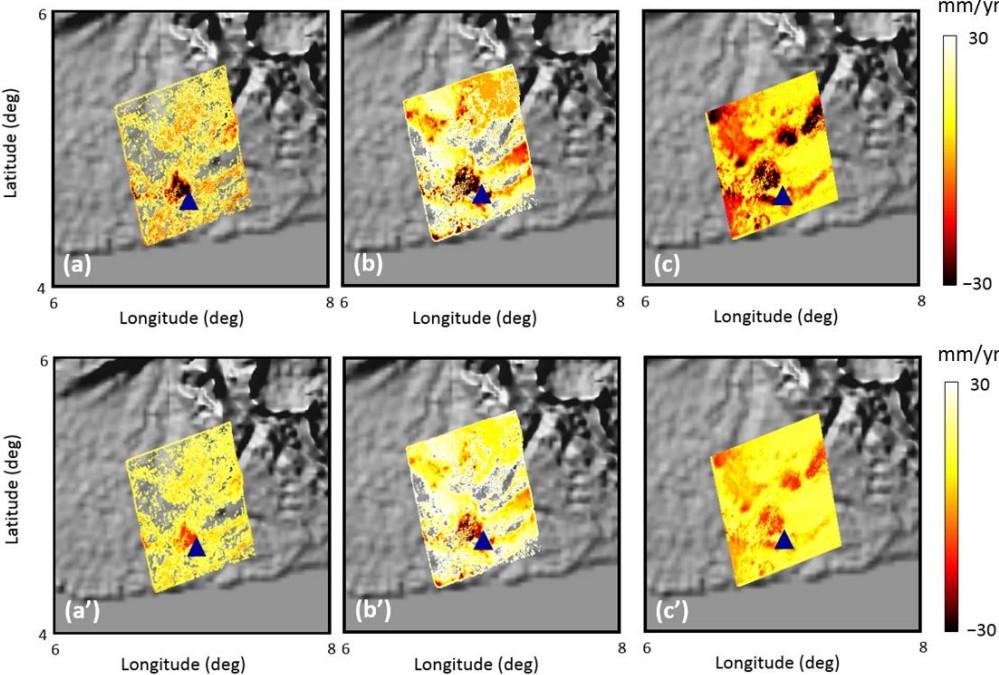

**Figure 7.** Obtained mean displacement rate from raw ((**a**) PSI, (**b**) SBAS, and (**c**) InSAR) and corrected ((**a′**) PSI, (**b′**) SBAS, and (**c′**) InSAR) interferograms.

It is evident that there is a subsidence signal in the center to the southwest and some portion of the east part of the study area. The subsidence rate is between −30 and 30 mm/yr, according to the calculation.

A comparison of the obtained deformation velocities from the GPS time series and SAR processing approaches' outputs can be seen in Figure 8 and Table 2. According to this comparison, the deformation accuracy of InSAR, PSI, and SBAS is improved by 58%,

42%, and 28% when using the AIM to consider APS correction. To explain more precisely how the percentages of improvement in each approach are created (Table 2), it should first be mentioned that the amount of deformation at the specific point near the GNSS station, which was created from different approaches, was calculated separately, and that the same calculations were repeated after implementing the AIM in each approach. A percentage difference between the results before and after the corrections was used to calculate the improvement in results.

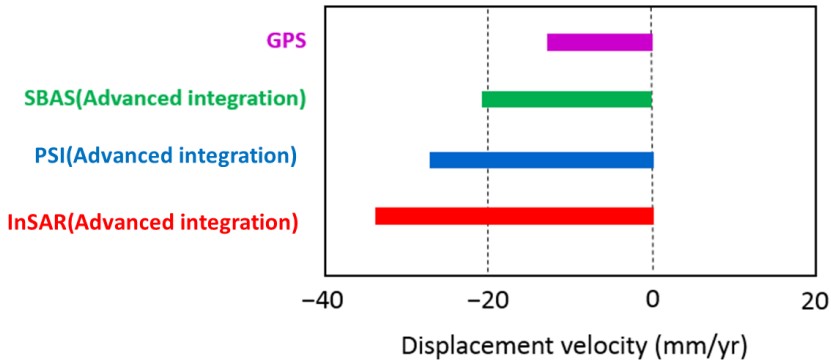

**Figure 8.** Comparison of obtained APS-corrected mean displacement rates with GNSS data.

**Table 2.** The percentage of improvement in each SAR processing approach via APS correction.

| SAR Processing Approaches at the Specific Point Near the GNSS Station | Velocity (mm/yr) before APS Correction | Velocity (mm/yr) after APS Correction | The Percentage of Improvement |
|---|---|---|---|
| SBAS | −27.7 | −20.3 | 28% |
| PSI | −47.0 | −27.3 | 42% |
| InSAR | −80.2 | −33.7 | 58% |

To make a meaningful comparison between the results, it is necessary to point out that the GPS observations have been refined and processed to eliminate noise.

After applying the AIM, the results mentioned below (Table 2, Figure 8) were obtained due to the comparison and improvement of the outputs. First, the SBAS method produces the best results in this study area because it involves the lowest amount of improvement, meaning that the troposphere delay decreases after SAR image processing in this technique.

Secondly, based on Figure 8, it can be stated that the SBAS method has had the best performance because its results are closer to GPS outputs compared to the other techniques. Using the AIM has caused the results of the SBAS algorithm to have the most negligible difference with the velocity obtained from validation data (the GNSS station) compared to other techniques. This difference is about 7 mm per year for GPS data. In addition, the InSAR method had the most negligible velocity accuracy. However, the use of the AIM has caused the results of this method to be closer to the results obtained from other approaches. Although the PSI algorithm utilizing the AIM has a closer velocity to the results of the SBAS method and GPS observations, it can be seen in Figure 7 that some part of the case study, which has deformation, was not shown in the PSI method. Based on this comparison, it can be concluded that the SBAS algorithm using the AIM has the highest accuracy in calculating the velocity compared to the other mentioned approaches. The obtained results are related to the velocity, so the differences between the approaches are significant and cannot be ignored.

According to the previous sections, there is only one GPS station in this study area. To validate the deformation in this study, the trend in the time series of vertical component displacements recorded between GPS stations over two years was compared with the results of two SAR data processing approaches, including SBAS and PSI. As can be seen in Figure 9, although the trend in displacements obtained from both the PSI and SBAS

techniques is similar to the trend in vertical displacements over time at the GPS station, as shown by the atmospheric correction method that the SBAS technique has the closest results to the GPS station, in this comparison the deformation process obtained from SBAS is entirely different from other processing approaches and is similar to the GPS station in terms of value and trend.

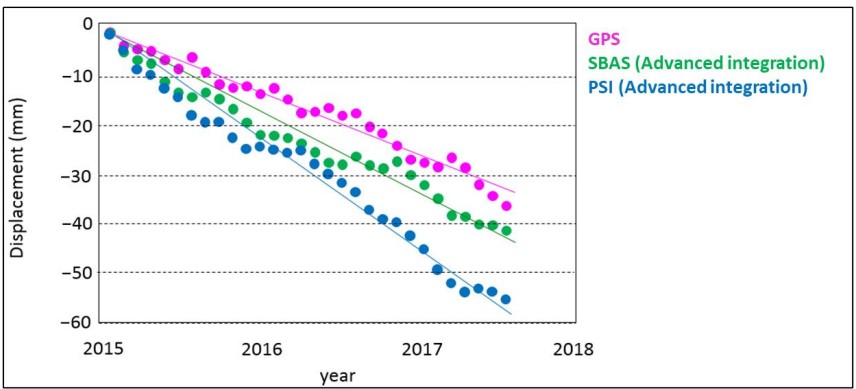

**Figure 9.** Comparison of the time series trend of the GNSS station with the time series of the PSI and SBAS techniques.

The purpose of preparing Figure 10 is to investigate and compare the behavior of the linear fitting of the GNSS station's vertical component with the deformation trend obtained from the SBAS and PSI techniques, which is mentioned in the previous figure (Figure 9) via the trend in the time series of groundwater levels at the location of the GNSS station, which was obtained from GRACE data. As shown in this figure, the downward trend and behavior of both graphs indicate subsidence in the region, and the slope of the line, determined by linear fitting, is −0.022 and −0.07, respectively, for each equation, confirming this. In addition to seasonal fluctuations in water level, subsidence is also quite visible in this time series.

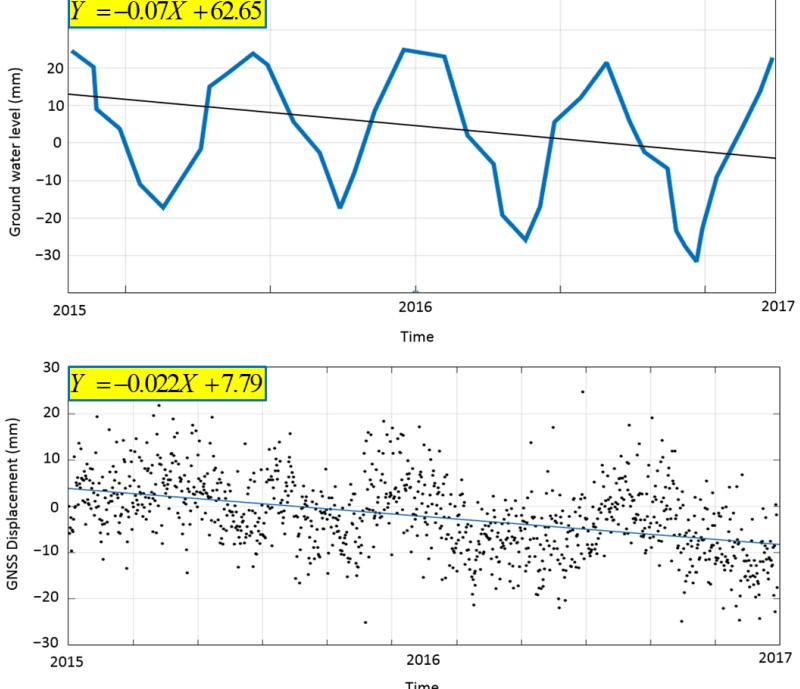

**Figure 10.** Comparison between the trend in the time series of groundwater levels obtained from GRACE data with linear fitting ((**above**): black line) and the time series trend in the vertical component of the GNSS station ((**below**): blue line).

*5.2. Classified Subsidence Signals*

Unsupervised Clustering

As mentioned in the Section 4, there is no label for SAR data. These methods have created MDRMs and classified them by considering centers close to each other; longitude, latitude, and velocity were calculated from three different SAR approaches. Therefore, MDRMs were prepared with several updates of each point according to nearing the centroids. The optimal and most corrected method was first selected with the Davies–Bouldin score criterion (Table 3) to evaluate the processing of implementing UCAs in APS-corrected data. The similarity measure table (Table 4) was prepared to evaluate the accuracy. In both tables (Tables 3 and 4), the results illustrate the fact that the SBAS technique with K-medians clustering gave us the best output (interval) for the deformation map, which had 91% accuracy with the usual clustering criteria; in this case, the standard deviation and equal interval methods. The map obtained from the SBAS technique with K-medians clustering shows that the mean displacement rate in the area in the center to the southwest of the case study is wholly correlated with the GPS-based maps of deformation. As can be seen in Figures 11–13, the SBAS technique is more matched to the mean deformation rate subsidence obtained from the GNSS station, and the subsidence rate map is between −25 and −10 mm/yr in the center to the southern part of the case study. Additionally, in the figures below, it is evident that the information obtained from the InSAR technique in the northwest, east, and central to east parts indicates subsidence of about 20 mm/year; based on the investigation and studying of the data obtained from the GNSS station as well as area of interest, these deformations are unrealistic. Moreover, as seen in Figure 9, the APS corrections in the SBAS method were more consistent with the GPS station, and unsupervised clustering methods also provided the best MDRMs with a reliable and proper numerical interval in this method.

**Table 3.** The Davies–Bouldin scores for the data obtained from the InSAR, SBAS, and PSI techniques.

| InSAR Processing Techniques with Three Clustering Methods | DBI |
| --- | --- |
| InSAR technique with fuzzy clustering | 0.535586 |
| PSI technique with fuzzy clustering | 0.454458 |
| SBAS technique with fuzzy clustering | 0.478788 |
| InSAR technique with K-means clustering | 0.513885 |
| PSI technique with K-means clustering | 0.443950 |
| SBAS technique with K-means clustering | 0.438856 |
| InSAR technique with K-medians clustering | 0.569536 |
| PSI technique with K-medians clustering | 0.493579 |
| **SBAS technique with K-medians clustering** | **0.397198** |

**Table 4.** The accuracy measures for each cluster are based on similarity measures.

| InSAR Processing Techniques with Three Clustering Methods | Accuracy |
| --- | --- |
| InSAR technique with fuzzy clustering | 0.8645 |
| PSI technique with fuzzy clustering | 0.8649 |
| SBAS technique with fuzzy clustering | 0.8954 |
| InSAR technique with K-means clustering | 0.8736 |
| PSI technique with K-means clustering | 0.8497 |
| SBAS technique with K-means clustering | 0.8984 |
| InSAR technique with K-medians clustering | 0.8796 |
| PSI technique with K-medians clustering | 0.8826 |
| **SBAS technique with K-medians clustering** | **0.9146** |

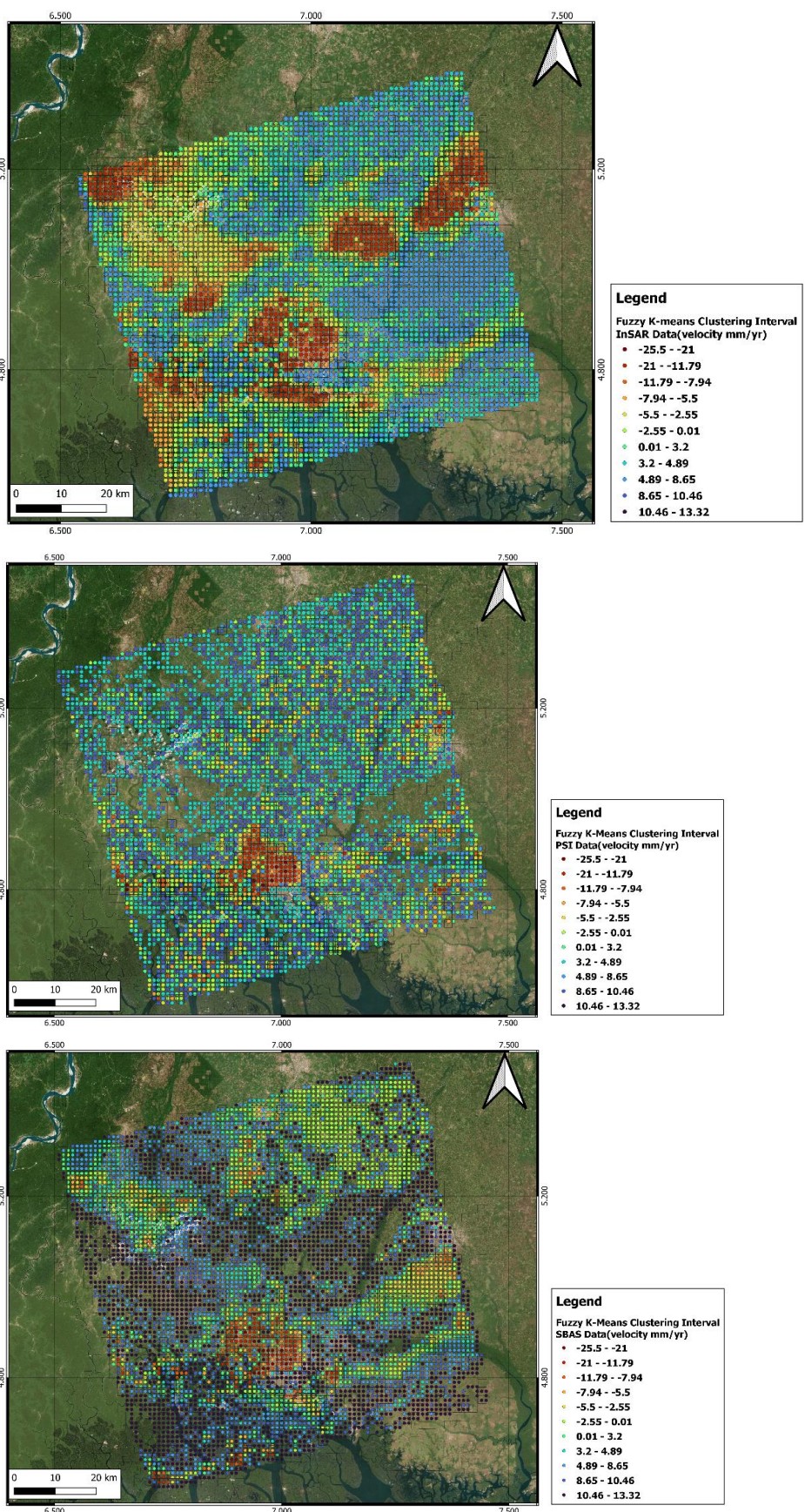

**Figure 11.** Obtained mean displacement rate maps from different interferometric techniques, including (**up**) InSAR, (**mid**) PSI, and (**down**) SBAS, by applying the fuzzy K-means clustering algorithm.

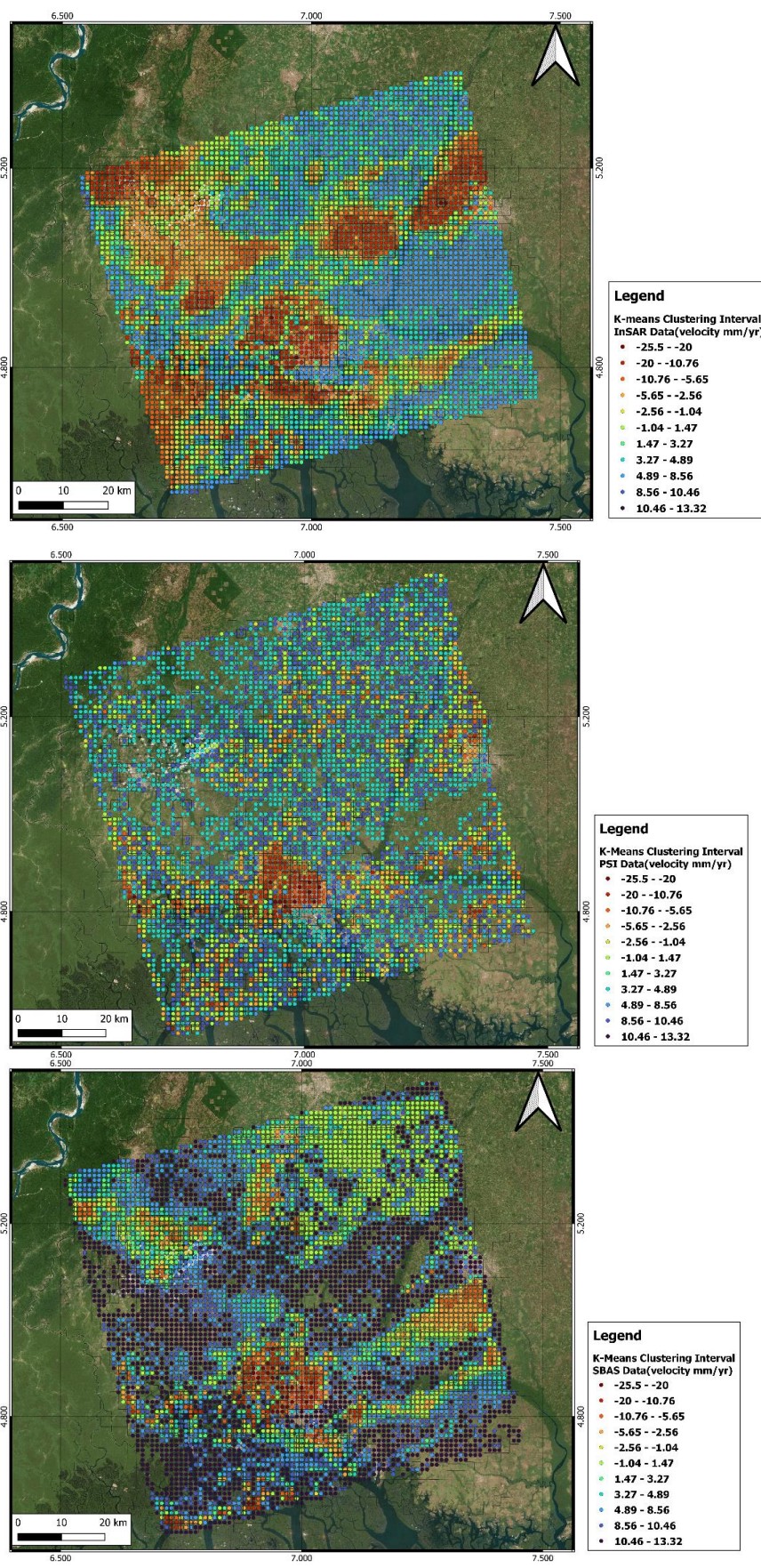

**Figure 12.** Obtained mean displacement rate maps from different interferometric techniques, including (**up**) InSAR, (**mid**) PSI, and (**down**) SBAS, by applying the K-means clustering algorithm.

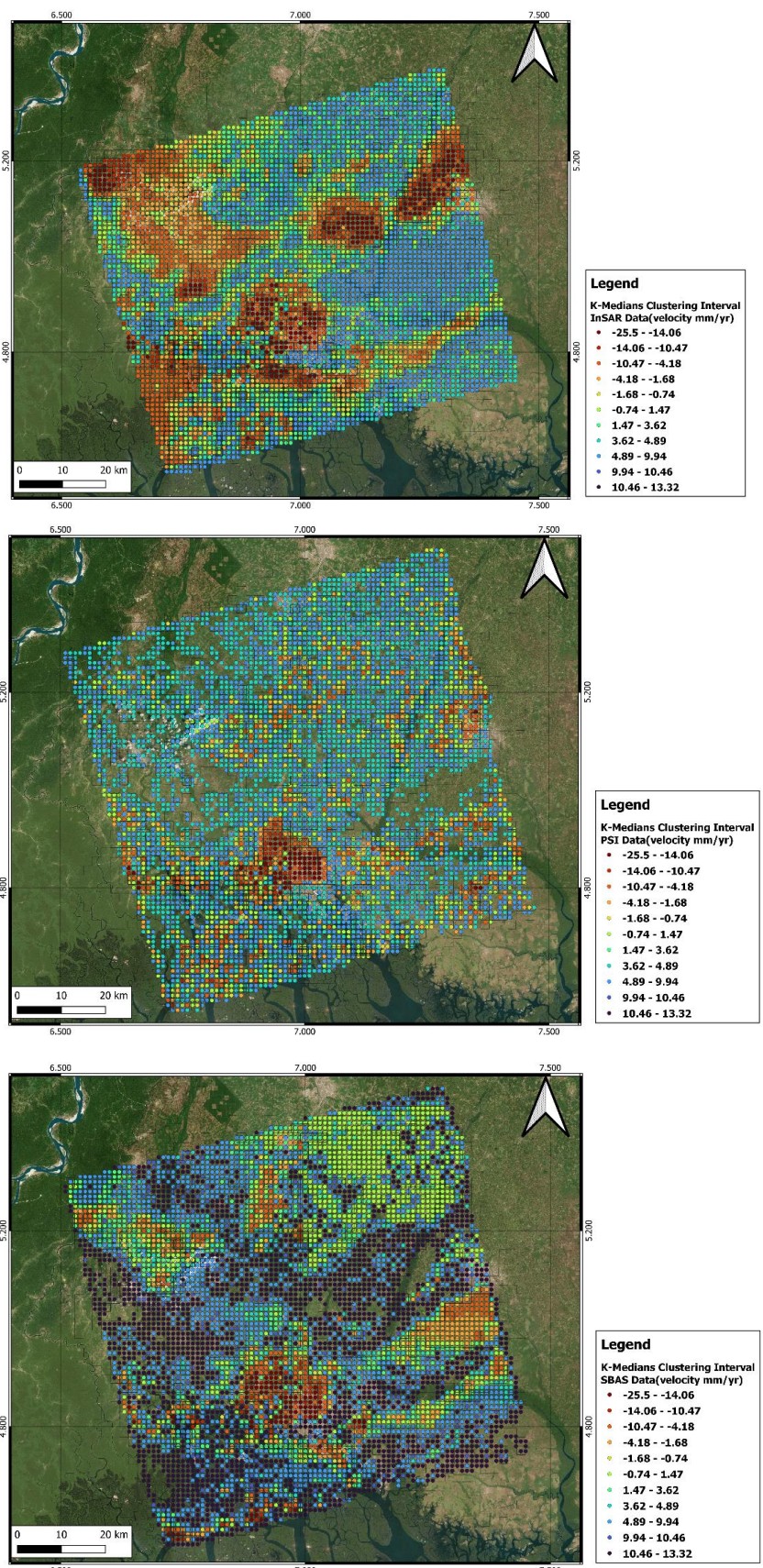

**Figure 13.** Obtained mean displacement rate maps from different interferometric techniques, including (**up**) InSAR, (**mid**) PSI, and (**down**) SBAS, by applying the K-medians clustering algorithm.

According to the Section 4, the Davies–Bouldin index and similarity measure were calculated to investigate the results of UCAs. The SBAS technique with K-medians clustering accurately illustrates the MDRM with the real interval for the determination of the subsidence and risk zone of the area. Tables 3 and 4 illustrate the accuracy of these methods for mapping mean deformation rates for the data obtained from the InSAR, SBAS, and PSI techniques.

The similarity measure method divides data from SAR processing techniques, such as InSAR, SBAS, and PSI, into a 20% test set and an 80% training set. The clustering methods mentioned above are used to label 80% of the data. Based on the similarity measure and the correlation distance between each cluster in the training set, the distance between each member of the test set is calculated. Then, the cluster with the minimum distance from the total distances of each test set with clusters 0, 1, 2, 3, and 4 is found and labeled to that member of the test set. To determine the best method, the results obtained by each tested member are compared with the labeling outputs determined by the general method of standard deviation [57]. As can be seen, the output of this evaluation is the same as the output of the DBI evaluation, which shows that the results are correct. The idea of this type of similarity measure comes from supervised machine learning and is used for the evaluation of UCAs. Table 4 shows the measures of accuracy for each cluster based on the formulae and explanations above, as well as illustrating the case study's more accurate subsidence and deformation.

In addition, UCAs use features such as latitude and longitude for the location of data, as well as velocity (mm/yr) and weights for area visualization, so it is expected that, after interpolation over the whole area of interest, the unsupervised (SBAS technique with K-medians clustering) maps will differ from the processed SAR images. The first step was to determine the average amount of deformation in the GNSS station with interpolation around the station by a radius of 500 m. After that, the latitude, longitude, and deformation rate of each pixel (mm/yr) were input into the unsupervised clustering algorithms according to the Section 4 of the study. Finally, an interpolation was conducted on the entire area, such that the pixels and deformation rate were continuously calculated based on the unsupervised classifications. The latitude and longitude of the GNSS station were used to calculate the amount of deformation in this station according to different clustering algorithms. It was found that using unsupervised clustering algorithms could help visualize and interpret the mean deformation rate map with 5.5% more accurate mean values from SAR image processing techniques, the vertical component of GNSS, and SBAS K-medians (Table 5).

**Table 5.** Percentage of similarity for deformation value obtained from selected clustering methods applied to results from the GNSS station.

| Method | Similarity with GNSS Station |
|---|---|
| SBAS | 89% |
| SBAS technique with K-medians clustering | 94.5% |

## 6. Discussion

To better understand the importance of using UCAs, two conventional and standard methods provided by software and packages for mapping and assessing SAR image outputs, such as standard division (STD) and equal interval, were implemented, and maps of the deformation of the area were created by using their interval. After evaluating the UCAs presented in Tables 3 and 4, including the DBI and similarity measures, the SBAS method was selected with the K-medians algorithm. Now we will compare the final output of this study with the two presented methods (Figure 14), which clearly demonstrate the advantages of using the algorithm of artificial intelligence to find the best interval for preparing MDRMs for interpreting the results. As evident in the figure, in the map related to the selection of the interval by standard division, the center towards the southwest, east,

and northeast, as well as parts of the northwest of the case study have undergone intense deformation. By looking at the map obtained by applying the equal interval method, we can see that the main area affected by deformation is the center to the southwest, without considering other sensitive zones summarized in the results. The output from the study, however, shows that the severe displacements are related to the center towards the southwest. The deformation has negatively affected the east, north, and northwest areas.

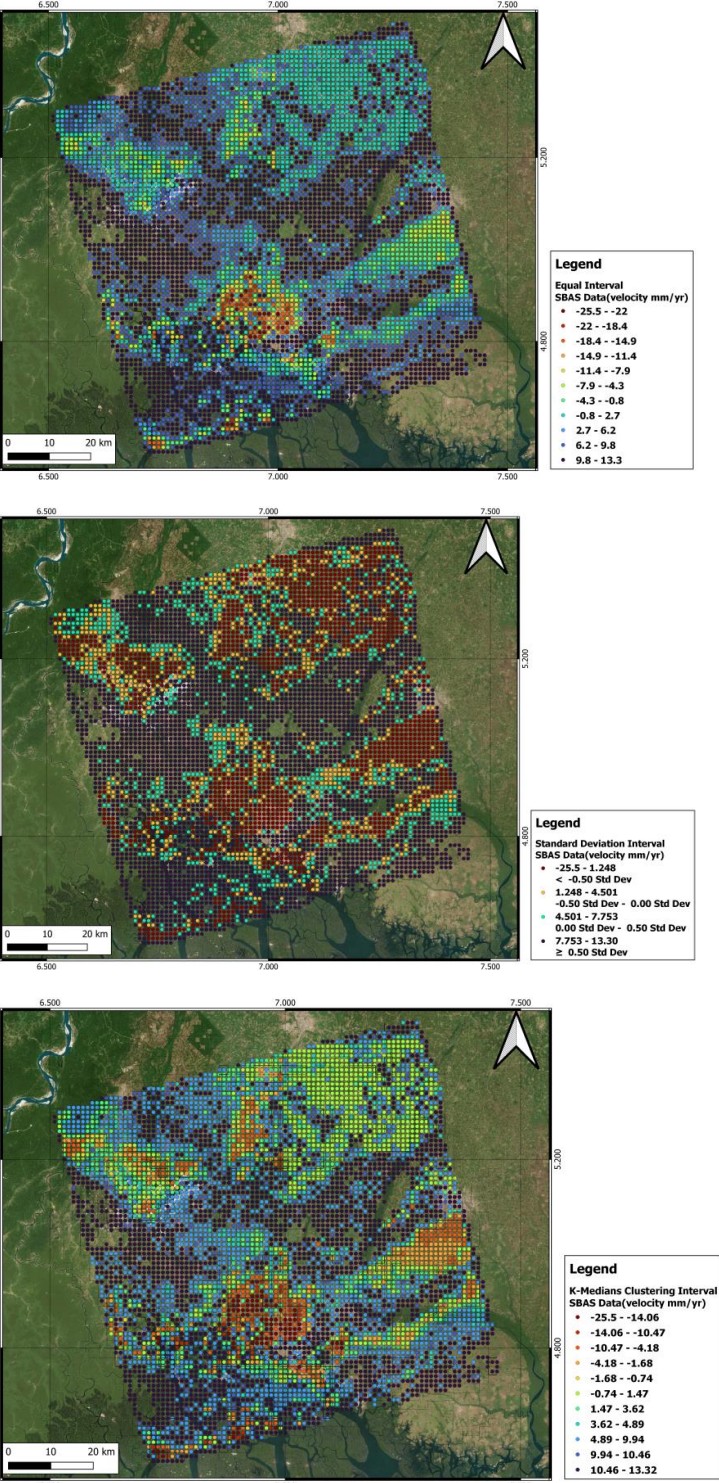

**Figure 14.** Comparison means displacement rate maps from SBAS techniques, including the (**up**) equal interval method, (**mid**) STD method, and (**down**) K-medians clustering algorithm.

In the results obtained from the InSAR method, notable subsidence signals are observed in the northwest and east of the study area. Many of these signals have disappeared from the map after applying the AIM. This indicates that failing to use the appropriate tropospheric correction method leads to incorrect perceptions of regional displacement signals.

In PSI velocity maps, subsidence signals can be seen in the center to southwest parts of the study area. Only one subsidence signal is observed near the GNSS station. Additionally, in other parts of the case study, there are no intense deformation signals.

Nevertheless, some SBAS-derived significant subsidence spots in the region's western part are suppressed when using the AIM to correct the APS. When using this method, the deformation has slightly more positive values in the northwestern part. SBAS seems to have shown better performance in suppressing the APS. After implementing UCAs and the union–intersection principle between intervals, the best MDRM was prepared, which was supported by a previous paper on the subsidence of this area [58]. Moreover, small areas with significant subsidence can be seen in the SBAS deformation model, with lesser significance in InSAR and PSI outputs.

## 7. Conclusions

Three UCAs were implemented in this paper to prepare MDRMs. As is known in the literature, atmospheric factors greatly influence interferometric chains, especially the area investigated in this work, in terms of water vapor pressure, air temperature, pressure, and other characteristics. Therefore, preparing APS corrections to enhance deformation signals is necessary for obtaining reliable results. In this regard, the results listed below were derived:

- The AIM for APS correction with ERA5 meteorological data significantly reduced the troposphere and atmospheric delays. One of this paper's most critical aspects is using the AIM for the first time in three different processing approaches, including SBAS, PSI, and InSAR.
- After implementing the APS correction method for the InSAR, PSI, and SBAS techniques, it was demonstrated that this correction method has improved by 58%, 42%, and 28% for them, respectively, which means that the SBAS technique has created interferograms with higher accuracy during processing in this study area. Additionally, the corrected result value of the SBAS method is consistent and more similar to the vertical component of the GNSS station.
- When the specialists start to process SAR data, they use the intervals of software or packages, either commercial or open source. It is important to emphasize that the MDRMs will change when we use different intervals in the case study. To illustrate reliable deformation parts of the case study, it is essential to identify the best and most accurate intervals based on the case study and deformation zone.
- After choosing the best method with which to find the most reliable and accurate intervals via similarity measures and the DBI, an evaluation of the deformations around the GNSS station within a radius of 500 m was performed. As stated in the Section 4, clustering methods were used to update the points according to latitude, longitude, and velocity. A subsequent analysis of the deformation of the SBAS technique results was conducted without considering the clustering methods, and the same analysis was performed for the SBAS K-medians. Finally, we can observe that the outputs provided by the point position update in clustering methods are 5.5% more accurate than the vertical displacements of the GNSS station.

**Author Contributions:** Conceptualization, M.A.K.; data curation, M.A.K. and L.G.; formal analysis, M.A.K., B.V., D.D.M. and S.H.-A.; methodology, M.A.K., D.D.M. and B.V.; software, M.A.K. and S.H.-A.; validation, D.C., D.D.M., L.G. and B.V.; writing—original draft preparation, M.A.K., L.G. and S.H.-A.; writing—review and editing, D.C., D.D.M. and B.V.; supervision, D.D.M. All authors have read and agreed to the published version of the manuscript.

**Funding:** The Ph.D. project is funded according to Art. 4 L.210/98 and the University Ph.D. regulations. The scientist responsible is Dr. Diego Di Martire.

**Data Availability Statement:** No new data were created or analyzed in this study. Data sharing is not applicable to this article.

**Acknowledgments:** The authors appreciate the ESA and ECMWF for providing radar acquisitions and ERA5 data. The authors thank Consorzio inter Universitario per la prevenzione dei Grandi Rischi (CUGRI) for providing technological support. Detailed and constructive reviews as well as editorial remarks are greatly appreciated.

**Conflicts of Interest:** The authors declare no conflict of interest.

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
