# Peer review of "Mapping of Mean Deformation Rates Based on APS-Corrected InSAR Data Using Unsupervised Clustering Algorithms"

_remotesensing, doi:10.3390/rs15020529_

Round 1

Reviewer 1 Report

In this paper, the APS correction method and three unsupervised clustering algorithms are incorporated to obtain more accurate subsidence and deformation. Experiments on different dataset are conducted to validate the effectiveness of different combinations. Finally, SBAS technique with K-Medians clustering is selected as the best choice for deformation map acquisition. However, some concerns are suggested below to improve the work.

1. The writing of the manuscript should be greatly checked and improved, especially the punctuation. For example, in the abstract the “;” should be change to “,” .

2. The full name should be given while the abbreviation appears in the manuscript for the first time. For example, SAR, PSI, SBAS has been introduced in the abstract and keywords without the full names.

3. The overall format of the article needs to be adjusted to meet the standard. Several formulas and words are misaligned. The text font and size are uniform.

4. The abstract should be carefully rearranged and polished up to highlight the innovation.

5. What are x and y in equation (8)? The explanations should be given.

6. Section 4 needs to be carefully revised and reorganized for introducing the work better. It is recommended to expand the introduction of the related basis of InSAR atmospheric phase model. Moreover, the innovation of this paper should be highlighted.

7. The core contribution of this paper seems to be the comparisons of the effectiveness of the existing method combinations. Hence, the manuscript is suggested to be reorganized.

Reviewer 2 Report

Review:

In this paper, three unsupervised clustering algorithms and advanced atmospheric phase screen (APS) correction integration method are used to obtain the ground deformation rate of InSAR. Atmospheric effect is indeed one of the main factors affecting the accuracy of InSAR deformation monitoring. It has important research value. However, there is no innovative research in this paper, only the conventional APS correction method is used for correction, and the clustering analysis of deformation rate has no practical value. Moreover, the writing level of this paper is poor, the logical expression is unclear, and the experimental results are not clearly expressed. Therefore, I don't think the requirements for publication have been met.

Detailed comments are as follows:

(1) Because the line numbers are not marked in the text, it brings great difficulties to peer review, and it is difficult to specify the line numbers during peer review.

(2) Groundwater level is mentioned in both the abstract and the introduction, especially in the introduction, which describes a lot of relevant contents of groundwater level. However, groundwater level is not the key content in actual research, and it is almost useless.

(3) It is not meaningful to cluster the deformation rate points obtained by InSAR.

(4) The principle in this paper is not enough to support the experimental results. For example, cluster analysis simply lists the formulas, but does not closely combine InSAR.

(5) The format of the text is confusing. Please check the formulas in the text, such as formulas (3), (4) and (6).

(6) Please modify the subtitle writing format in the text.

(7) What do the sub-tables (a) and (b) in Figure 5 mean?

(8) The displacement velocity fields obtained by PSI, SBAS and InSAR methods are shown in Figure 6. What does InSAR method mean here? Aren't PSI and SBAS InSAR methods?

(9) There is a big problem in Figure 7, which describes "The deformation accuracy of InSAR, PSI, and SBAS is improved by 58%, 42%, And 28% when using an advanced integration method to consider APS. ",how can we calculate the 58%, 42%, and 28%% here? Since it is an increase, we should give the deformation rate before using the advanced integration method.

(10) the article describes "As it is evident in this figure, the groundwater level trend is similar to the vertical component of GPS and the deformation trend obtained from the SBAS and PSI technique, Which gives a good validation for the accuracy of the SAR processing. ",but the vertical component of GPS and the deformation trend obtained by SBAS and PSI technologies are not shown in Figure 9.

(11) The format of references does not meet the requirements of journals.

Reviewer 3 Report

The objective of this paper is incorporating the APS correction method, in both InSAR and A-DInSAR processing techniques, which will eventually result in the estimation of accurate mean deformation rate maps for the Port Harcourt area (Nigeria).

This is an interesting paper, which is properly structured, as it contains all necessary sections (Introduction, Study area, Datasets, Methods, Results and Validation, Discussion, Conclusions). In addition, the “Methods” and “Results and Validation” sections are divided into multiple sub-sections, providing additional details. Regarding the mathematical analysis, explained in the “Methods” section, it is valid and consistent with the extracted results, while Figures and Tables are consistent with the analysis, described in the text, respectively. However, some changes should be performed, which will overall improve the paper:

Abstract: The abstract should be more concise and include the most significant information. In its current form, more details are provided, which should be removed and placed in the manuscript main body. Please, modify.

“…caused by groundwater harvesting”: This part is lacking bibliographic references. Some indicative, recent and relative references, which can be optionally cited are: 1. He, Z., Chen, T., Wang, M., Li, Y., 2020. Multi-Segment Rupture Model of the 2016 Kumamoto Earthquake Revealed by InSAR and GPS Data. Remote Sens (Basel) 12, 3721. https://doi.org/10.3390/rs12223721, 2. Ghorbani, Z., Khosravi, A., Maghsoudi, Y., Mojtahedi, F.F., Javadnia, E., Nazari, A., 2022. Use of InSAR data for measuring land subsidence induced by groundwater withdrawal and climate change in Ardabil Plain, Iran. Sci Rep 12, 13998. https://doi.org/10.1038/s41598-022-17438-y, 3. Lazos, I., Papanikolaou, I., Sboras, S., Foumelis, M., Pikridas, C., 2022. Geodetic Upper Crust Deformation Based on Primary GNSS and INSAR Data in the Strymon Basin, Northern Greece - Correlation with Active Faults. Applied Sciences 12, 9391. https://doi.org/10.3390/app12189391, 4. Yalvac, S., 2020. Validating InSAR-SBAS results by means of different GNSS analysis techniques in medium- and high-grade deformation areas. Environ Monit Assess 192, 120. https://doi.org/10.1007/s10661-019-8009-8. Please, include relative references in this part.

Study area: Please, add a brief paragraph, in which the geological regime and the major tectonic structures will be described. It can be placed before the litho-stratigraphic analysis. Please apply.

Conclusions: Please, modify the “Conclusions” section. In its current form it resembles the abstract. The most significant findings of your research should be highlighted and briefly described. Numbering of the concluding remarks can be performed. Please, apply.

Round 2

Reviewer 1 Report

The authors have made great improvements and I suggested that the manuscript could be accepted after minor revision.

1. Please pay attention to the English writing and the format.

2. The introduction could be refined and the key issue could be made in brief.

Author Response

Thank you for your insightful comments, and it was a pleasure working with the reviewers. As a result of the reviewers' suggestions, all changes have been incorporated. Changes have been highlighted with Track Changes in the manuscript. The Word File is a point-by-point response.

Reviewer 2 Report

Although most of my suggestions have been revised and explained in the revised draft, the author mainly focuses on the format and text description, and its innovation, applicability and result display effect have not been significantly improved. In addition, I have made an overall evaluation of the paper in the first review process, and have given some suggestions for rejecting the manuscript, so the final result still needs to be decided by the relevant editors.

Author Response

Thank you for your insightful comments, and it was a pleasure working with the reviewers. As a result of the reviewers' suggestions, all changes have been incorporated. Changes have been highlighted with Track Changes in the manuscript. Please see the word file.
